# New Psychometric Evidence of the Life Satisfaction Scale in Older Adults: An Exploratory Graph Analysis Approach

**DOI:** 10.3390/geriatrics9050111

**Published:** 2024-09-02

**Authors:** Julio Dominguez-Vergara, Brigitte Aguilar-Salcedo, Rita Orihuela-Anaya, José Villanueva-Alvarado

**Affiliations:** 1Research Direction, Universidad Tecnológica del Perú, Lima 15046, Peru; 2Research, Science and Technology Unit (UICT), Faculty of Psychology, Universidad Peruana Cayetano Heredia, Lima 15074, Peru; brigitte.aguilar.s@upch.pe (B.A.-S.); rita.orihuela@upch.pe (R.O.-A.); 3Academic Department of Psychology, Pontificia Universidad Católica del Perú (PUCP), Lima 15088, Peru; villanueva.alvarado@pucp.edu.pe

**Keywords:** life satisfaction, older adults, psychometrics, exploratory graph analysis, Peru

## Abstract

The objective of the present study was to analyze the psychometric properties of a life satisfaction scale in older Peruvian adults using an exploratory graph analysis (EGA) approach. A total of 407 older adults aged between 60 and 95 years (M = 69.5; SD = 6.7) from three comprehensive elderly care centers (CIAMs) in Lima, Peru, were recruited. A non-probabilistic convenience sampling was used. The Satisfaction with Life Scale (SWLS) was analyzed using EGA with the Gaussian GLASSO model to assess its dimensionality and structural consistency. The relationship with other variables was analyzed using scales such as the GAD-7 and PHQ-9. The network structure of the SWLS indicates a single dimension. Additionally, network loadings (nodes) were examined, showing high values (>0.35) for most items except item 1, which had a moderate loading (>0.25). Structural reliability showed that a single dimension was identified 100% of the time. The post hoc CFA considering the unidimensional network structure obtained through EGA showed satisfactory fit (χ2/df = 3.48, CFI = 0.96, TLI = 0.92, SRMR = 0.02, RMSEA = 0.07 [90% CI 0.05, 0.08]). Finally, internal consistency reliability was acceptable (ω = 0.92). The SWLS measure is robust and consistent. These findings are a valuable reference for advancing research on aging in Peru, as they provide a practical, valid, and reliable measure.

## 1. Introduction

The increase in life expectancy among older adults (OAs) has driven the development of new social, economic, and health policies aimed at improving quality of life [1] and has also led to an increase in scientific research [2]. While demographic projections show a rise in the average life expectancy of older individuals, healthy life expectancy refers to the number of years of good health that an individual can expect to live at a given age [3]. In Latin America, the United Nations [4] reports that in 2022, the older adult population reached 88.6 million, representing 13.4% of the total population, with projections estimating an increase to 16.5% by 2030.

Peru also reflects this demographic trend; according to the National Institute of Statistics and Informatics [INEI] [5], people over 60 years old make up 13.6% of the country’s population, and it is estimated that by 2070, the proportion of older adults will reach 30.1%. Regarding social factors, 33.8% of working older adults are affiliated with a pension system, and 26.8% of Peruvian households have at least one older adult as the head of the household [5]. Therefore, the quality of life for older adults requires continued employment until very advanced ages, and the quality of support networks is also relevant as it can be a key factor in life satisfaction. In health, despite 92.1% of people over 60 years old having some form of health insurance, there is a decreasing trend in seeking medical care due to high demand and inefficient care processes [6]. These issues have led older adults to experience depression, anxiety, loneliness, and feelings of abandonment, significantly affecting their quality of life [7]. Among cultural factors, older adults value functional health through independence and the ability to perform daily activities; they also appreciate active participation in religious, leisure, and social integration activities [8].

These factors present challenges for social and health services in the pursuit of a satisfying life [9]. Therefore, improving the health and life satisfaction of older adults becomes crucial, as being content with one’s current life is considered an indicator of healthy aging [10]. Additionally, life satisfaction, being a subjective assessment, is susceptible to contextual changes and is influenced by the perceptions and interpretations of older adults, relating to their health and the social and economic conditions of their environment [11]. In this regard, Diener [12] emphasizes the importance of the global cognitive evaluation that older adults make of their own lives. Various studies indicate a positive relationship between life satisfaction and subjective health, as well as its impact on self-esteem, motivation, overall health, and better coping strategies [13,14]. In this context, life satisfaction becomes relevant because it helps to demystify the stigmas about older adults.

Various studies have demonstrated that physical activity, mood, and family support influence life satisfaction [15]. Additionally, other research has shown that factors such as living conditions [16], neighborhood environment [17], technology use [18,19,20], healthy eating [21,22], transportation [23], and social service [24] positively influence life satisfaction among older adults.

Therefore, there is a need to assess life satisfaction using instruments that ensure adequate validity and reliability. The Satisfaction with Life Scale (SWLS) was developed by Diener et al. [25] and is the most widely used measure in various studies. This measure can be applied to different age groups, has been translated into more than 30 languages, and demonstrates good psychometric evidence through reliability (internal consistency and test-retest) and validity evidence (content analysis, internal structure, and convergent validity) [26]. In Latin America, the SWLS has been validated in older adults in countries such as Mexico [27], Peru [28], Chile, and Ecuador [29], focusing on exploratory, confirmatory, and invariance methods. Given that the SWLS is widely used, new validations considering cultural variation are necessary, as differences and similarities in life satisfaction judgments emerge over time due to sociocultural influences [30].

In the last decade, network models have emerged as an alternative for exploring data structures; these models also complement existing latent variable techniques such as multidimensional scaling and exploratory factor analysis [31]. In contrast to latent variable models derived from classical test theory (CTT), the determination of the internal structure of latent factors can lead to a lack of consensus in the definition and interpretation of the obtained factors [32]. An innovative way to address the relationships between items is through network analysis [33,34]. One of the challenges is the ability to visualize relationships in a diagram composed of nodes (items) and edges (partial correlations); additionally, the thickness of the edges allows for the examination of the strength of the relationships [35]. Exploratory graph analysis (EGA) is combined with a set of weighted networks [36], enabling the examination of network loadings in the node diagram, structural consistency, and facet detection algorithms [34]. In this way, EGA allows for the immediate interpretation of elements belonging to each factor through the network graph using colors, and influential relationships between items and dimensions can be observed without the need to make decisions about the type of rotation to use for the factor structure [37]. Therefore, EGA is a useful tool for exploring the factor structure and item interactions of the SWLS.

The main objective of this study is to explore the factor structure of the SWLS in older Peruvian adults using the EGA methodology. Although previous studies have analyzed the factor structure of the SWLS, these have been conducted in university samples, considering the instrument’s “free domain” nature [38,39]. The SWLS has been used in more than 4000 studies to assess an individual‘s overall evaluation of their own life. This instrument is highly relevant and applicable, as the information provided through exploratory graph analysis (EGA) can have significant implications for clinical practice with older adults. Understanding the dimensionality of the SWLS ensures that the scores derived from a single measure are valid and useful for assessing life satisfaction. Therefore, the SWLS can be a valuable tool for professionals in psychology, psychiatry, and geriatrics, enabling specific screening in mental health settings.

## 2. Materials and Methods

### 2.1. Design

This study is instrumental in nature [40], as it examined the internal structure of the SWLS with the aim of evaluating its validity and reliability in older Peruvian adults.

### 2.2. Participants

A total of 407 older adults, aged between 60 and 95 years (M = 69.5; SD = 6.7), were recruited from three comprehensive senior centers (CIAMs) in Lima, Peru. The selection was carried out through non-probabilistic purposive sampling. Among the advantages of this type of sampling are the adaptability of the design when the sample is specific and difficult to reach, the estimations can be sufficiently accurate if applied correctly, and it is useful for obtaining preliminary data. However, the disadvantages of non-probabilistic sampling, particularly concerning representativeness and generalizability, must be noted [41]. The details of the sociodemographic data are presented in Table 1.

### 2.3. Instruments

Data Sheet

The data sheet includes questions about age, gender, educational level, living arrangement, employment status, and physical activity practice.

Diener’s Satisfaction with Life Scale (SWLS, Diener et al., 1985 [25])

This is a self-report scale composed of five items that assess general life satisfaction. For this study, the Spanish version by Atienza et al. [42] was used. Response alternatives are on a five-point Likert scale ranging from ‘strongly disagree’ (1) to ‘strongly agree’ (5), with higher scores indicating greater satisfaction. This scale was validated in the Peruvian context in a sample of older adults by Caycho-Rodríguez et al. [28]. Among its psychometric properties, the SWLS showed a unidimensional structure with acceptable fit indices (χ2 = 10.960, df = 5, *p* = 0.05, χ2/df = 2.192, GFI = 0.983, CFI = 0.994, NFI = 0.988; RMSEA = 0.071 [90% CI 0.000, 0.129] and SRMR = 0.013) and acceptable reliability (ω = 0.93).

Measures used for evidence are based on relationships with other variables.

Patient Health Questionnaire (PHQ-9)

The PHQ-9 consists of nine items that reflect depressive symptoms, evaluated over the past two weeks with Likert-type options (0 = not at all, 1 = several days, 2 = more than half the days, 3 = nearly every day). The total score ranges from 0 to 27. Validated in the Peruvian population, the PHQ-9 shows a unidimensional model with good fit indicators (CFI = 0.99; TLI = 0.987; SRMR = 0.048; RMSEA = 0.071) and adequate reliability (ω = 0.861) [43]. For this study, reliability was assessed using the omega coefficient, yielding a value of 0.89.

Generalized Anxiety Disorder Scale (GAD-7)

This is a widely used screening measure to assess generalized anxiety. It contains seven items, and respondents are asked to rate the frequency of their anxiety symptoms over the past two weeks. The GAD-7 response options range from 0 (not at all) to 3 (nearly every day). The GAD-7 was validated by Franco-Jiménez and Nuñez-Magallanes [44], showing a unidimensional model with good indices (χ2 = 31.717, CFI = 0.995, TLI = 0.992, RMSEA = 0.056, SRMR = 0.026) and high reliability (ω = 0.92). For this research, reliability was calculated using the omega coefficient, achieving a value of 0.92.

### 2.4. Data Collection

Data collection was conducted between May and September 2023. To identify the participants, first, contact was made with four comprehensive senior centers (CIAMs) in the districts of Lima. Second, the center coordinators were contacted to inform them about this study and to schedule potential dates for data collection. Third, the researchers attended the CIAMs during the seniors’ activity sessions and consulted them in advance about their availability to participate in this study. Fourth, informed consent was provided to the participants, outlining this study’s objectives, confidentiality, anonymity, and voluntary participation. Fifth, once the older adults agreed to participate, they completed the questionnaires individually. For those who were illiterate, the statements and response options were read aloud to ensure accurate completion of the instruments, thereby avoiding biases related to reading and comprehension difficulties.

### 2.5. Data Analysis

A descriptive analysis of the SWLS was performed, obtaining measures of mean, standard deviation, skewness, and kurtosis. Additionally, response rates per item were calculated in percentages (%), considering the ordinal nature of the variable. The dimensionality of the SWLS was verified using the Gaussian GLASSO model through exploratory graph analysis (EGA) with the EGAnet library [45]. Network loadings were also calculated using the “net.loads” function, where small (0.15), moderate (0.25), and large (0.35) network values were considered [46]. Structural consistency was examined using the “bootEGA” function, which extracts dimensions derived from EGA through a Bootstrap simulator with 1000 replications. Item stability is explained by the number of times it replicates in the same dimension. A minimum threshold of 75% was used to assess the consistency and stability of the SWLS items [47].

Once the factorial structure was obtained through EGA, a post hoc confirmatory factor analysis (CFA) was performed. For the analysis of the factorial model, the weighted least squares mean and variance adjusted (WLSMV) estimation method was used, which is suitable for items with ordinal characteristics. Model fit indices were examined using the chi-square test (χ2), the root mean square error of approximation (RMSEA) with 90% confidence intervals, the standardized root mean square residual (SRMR), the comparative fit index (CFI), and the Tucker–Lewis index (TLI). For model fit evaluation, values less than 0.08 for RMSEA and SRMR indices and values greater than 0.90 for CFI and TLI were considered. Additionally, factor loadings were calculated, all of which were greater than 0.50 for each item.

The estimation of model reliability was obtained using McDonald’s Omega coefficient (ω) with 95% confidence intervals, considering acceptable values greater than 0.80.

Statistical analysis was performed using the RStudio environment [48].

## 3. Results

### 3.1. Preliminary Analysis of the SWLS

Descriptive measures were calculated, finding that item 3 (“I am satisfied with my life”) had the highest mean (M = 3.70), while item 2 (“The conditions of my life are excellent”) had the lowest mean value (M = 3.51). The standard deviation was higher for item 3 (SD = 1.27) and lower for item 2 (SD = 1.12). Skewness and kurtosis coefficients for all items showed values exceeding ± 1.5, suggesting that the data do not follow a normal distribution [49]. Item–test correlations exceeded the criterion of 0.20 for all five SWLS items [50]. Response percentages for the items demonstrated a high agreement tendency in option 4 (“agree”) (view Table 2).

### 3.2. Exploratory Graph Analysis

Figure 1 shows the network structure of the SWLS items using EGA, where a single dimension is evident. Additionally, network loadings (nodes) were examined, with high values (>0.35) observed in most items, except for item 1, which presented a moderate loading (>0.25). Regarding structural consistency, it was evidenced that 100% of the time, a single dimension was identified; this is verified through the stability graph, where the items were systematically identified in a single community.

### 3.3. Post Hoc Confirmatory Factor Analysis and Relationship with Other Variables

A post hoc confirmatory factor analysis (CFA) was conducted, considering the network structure obtained through EGA. The unidimensional structure demonstrated satisfactory fit (χ2/df = 3.48, CFI = 0.96, TLI = 0.92, SRMR = 0.02, RMSEA = 0.07 [90% CI 0.05, 0.08]). Factor loadings were acceptable (λ > 0.5), ensuring that the items adequately represented the construct (Figure 2). Additionally, negative correlations were found between life satisfaction scores and anxiety (r = −0.144) and depression (r = −0.129).

Finally, reliability through internal consistency using McDonald’s omega coefficient was acceptable (ω = 0.92, 95% CI [0.91–0.94]).

## 4. Discussion

The present study aimed to validate the SWLS using a network model in older Peruvian adults. Among the findings, it was evidenced that the SWLS presents adequate evidence concerning its reliability, internal structure, and relationship with variables (anxiety and depression). Thus, the results support the unidimensional structure through EGA with precision and stability.

According to various psychometric studies, the SWLS has been validated using different factor and invariance methods in different samples, including adolescents [51], university students [38,52], clinical samples [53], and community samples [54]. However, the SWLS has not been analyzed in older adults through a psychometric network approach. Therefore, network models have proven to be superior to traditional factor analysis techniques [55,56], especially highlighting the absence of factor rotation and their intuitive nature.

EGA was used to examine the internal structure to evaluate the dimensionality and determine the number of factors present [47,57]. Thus, the unidimensional structure of the SWLS is consistent with the original model and supported by various studies with older adults [27,29,58,59,60]. Regarding connections within the network, item 3 (SWLS3: “I am satisfied with my life”) showed the strongest node loading in relation to the direct measure of life satisfaction. The SWLS items presented adequate loadings, indicating that they are directly linked to life satisfaction and suggesting a reciprocal cause–effect relationship between the network attributes [61]. Following the exploratory graph analysis, a post hoc confirmatory factor analysis of the SWLS was conducted, obtaining acceptable indices, thus validating a unidimensional structure that functions in older adults.

Reliability was determined through structural consistency, verifying that all data were systematically organized into a single dimension from replications. These results demonstrate the stability of the reciprocal variance between items. However, no previous study has analyzed the SWLS in older adults from a network analysis approach, but only under the framework of classical test theory. Adequate internal consistency values have been found through the estimation of Cronbach’s alpha coefficients [27,29] and Omega [28]. Despite this, the use of the network model is preferred due to its incompatibility with calculation, as common covariances are eliminated, valuing item correlations; moreover, internal consistency measures do not adequately report whether items remain through a unidimensional factor in multidimensional models [34]. In addition, this study also thoroughly analyzed internal consistency reliability, obtaining acceptable measures (ω = 0.92) for the SWLS.

The behavior of the SWLS with other measures found negative correlations with anxiety and depression. These results are consistent with previous studies that similarly report this association between the aforementioned variables [62,63]. This explains that the coexistence of anxiety symptoms can cause cognitive maladaptation, feelings of loneliness [64], and a decrease in quality of life [65], significantly affecting life satisfaction in older adults [66].

Among the practical implications, the use of the EGA methodology that supports the unidimensional determination of the SWLS stands out. In this way, this procedure contributes to the breadth of knowledge, as the EGA managed to identify a single dimension impartially and in congruence with what was analyzed in the literature. These findings offer a clear scope of the relationships of the items with the SWLS dimension, as well as how they position themselves within a single community. On the other hand, the internal structure of the SWLS conceptually contributes to the study of life satisfaction in samples of older adults in a Peruvian context. The use of the instrument will be useful in specific areas that support the accuracy of their evaluations, such as psychology, geriatrics, and medicine. The exploration of a single dimension is structurally in line with what is reported through various psychometric evaluation studies, with the SWLS functioning in various sociocultural contexts. Therefore, having a practical, brief, simple, and self-report measure will allow incorporating personal data on life satisfaction.

Despite the important results, this study has some limitations. First, although the sample size was moderate, it could not be subdivided to perform both validation procedures (EGA and CFA), so a post hoc CFA was conducted. For future research, it is suggested to adjust the unidimensional model of the SWLS using CFA with a similar sample. Second, the older adults recruited in comprehensive senior centers were selected through convenience sampling; therefore, they do not necessarily constitute an adequate representation of older adults in Peru. Third, the cross-sectional study does not allow for the estimation of temporal or directional relationships, limiting conclusions about causal relationships between the SWLS, depression, and anxiety. Fourth, due to non-probabilistic sampling, the sample was predominantly composed of women. Finally, self-report measures can introduce social desirability bias, as participants might present a positive image of their lives.

## 5. Conclusions

In conclusion, the SWLS shows satisfactory results in terms of validity and reliability, making it a suitable instrument for use and consolidation in future psychometric studies as a tool for measuring life satisfaction in older adults.

## Figures and Tables

**Figure 1 geriatrics-09-00111-f001:**
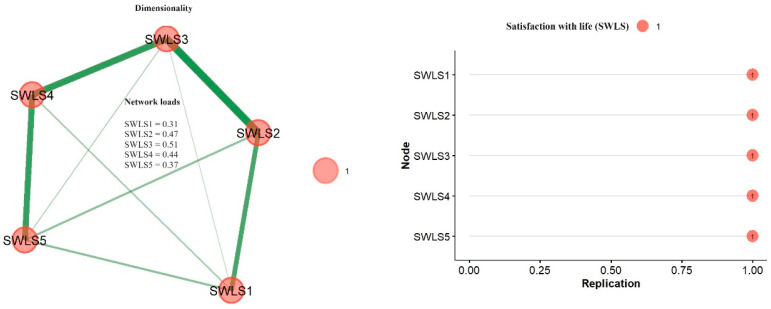
Dimensionality and structural stability of items.

**Figure 2 geriatrics-09-00111-f002:**
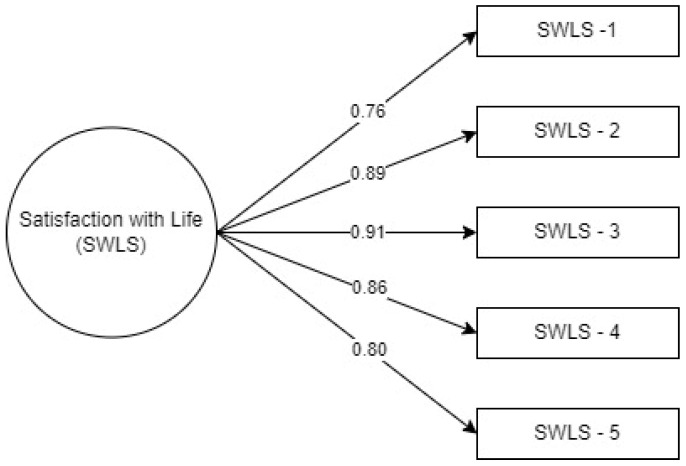
Factor structure of the SWLS.

**Table 1 geriatrics-09-00111-t001:** Characteristics of older adults (n = 407).

Variable	Category	Frequency	%
Age (M ± SD)		69.5 ± 6.7	
Sex	Male	160	39.3
	Female	247	60.7
Living Arrangement	Lives alone	35	8.6
	Lives with spouse	106	26
	Lives with children	120	29.5
	Lives with spouse and children	112	27.5
Marital Status	Single	40	9.8
	Married	227	55.8
	Cohabiting	26	6.4
	Divorced/separated	35	8.6
	Widowed	79	19.4
Educational Level	No education	8	2
	With basic education	181	44.4
	Higher education	122	30
Employment	Yes	103	25.3
	No	304	74.7
Physical Activity	Never	16	3.9
	Rarely	78	19.2
	Sometimes	177	43.5
	Frequently	108	26.5
	Very frequently	28	6.9

Note: M = mean; SD = standard deviation.

**Table 2 geriatrics-09-00111-t002:** Descriptive statistics, item–test correlation, and response percentages (n = 407).

SWLS Items	M	SD	g1	g2	Item–Test Correlation	Responses (%)				
						1	2	3	4	5
SWLS—1	3.47	1.06	−0.71	−0.01	0.74	7	10	25	45	13
SWLS—2	3.51	1.12	−0.81	−0.07	0.84	9	10	19	47	15
SWLS—3	3.70	1.27	−0.85	−0.27	0.85	11	6	19	33	32
SWLS—4	3.69	1.17	−0.76	−0.25	0.83	7	11	17	38	27
SWLS—5	3.54	1.25	−0.63	−0.56	0.78	10	10	21	33	25

Note: M = mean; SD = standard deviation; g1 = skewness; g2 = kurtosis; 1 = strongly disagree; 2 = disagree; 3 = neither agree nor disagree; 4 = agree; 5 = strongly agree; SWLS—1 = in most respects, my life is as I want it to be; SWLS—2 = the conditions of my life are good; SWLS—3 = I am satisfied with my life; SWLS—4 = so far I have gotten the important things I want in life; SWLS—5 = if I could live my life over, I would repeat it just the same way it has been.

## Data Availability

The data shown can be accessed by contacting the corresponding author and are available for academic and research purposes.

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
