# Peer review of "New Psychometric Evidence of the Life Satisfaction Scale in Older Adults: An Exploratory Graph Analysis Approach"

_geriatrics, 2024, doi:10.3390/geriatrics9050111_

Round 1

Reviewer 1 Report

Comments and Suggestions for Authors

The purpose of this article is to analyze the psychometric properties of the Satisfaction with Life Scale (SWLS) among the elderly population in Peru using Exploratory Graph Analysis (EGA) methods, which holds academic value for understanding healthy aging. Of course, the article has areas open for discussion, and here are my preliminary thoughts after reading it, for the author's consideration.

Firstly, the context of Latin American aging:

1With the increasing elderly population in Latin America, why has the focus on elderly life satisfaction become crucial?

2In studying life satisfaction among elderly in Peru, what specific social and cultural factors need consideration?

3What is the current research status regarding elderly quality of life and satisfaction in Latin America, particularly in Peru?

Secondly, methodological innovations:

1What innovations does Exploratory Graph Analysis (EGA) offer compared to traditional factor analysis? How does it enhance the understanding of the structure of the SWLS?

2Why choose EGA over traditional factor analysis to analyze the dimensional structure of SWLS?

3What challenges did the article encounter in applying the EGA method to analyze SWLS?

Thirdly, data availability:

1The study recruited 407 elderly aged 60 to 95 using non-probability convenience sampling. What are the advantages and disadvantages of this method?

2How was the accuracy and reliability of data ensured when collecting data using SWLS, PHQ-9, and GAD-7 scales

3What was the process of data collection like in Peru for such a survey study? Were there difficulties in obtaining data?

Finally, significance of the study:

1The study shows that SWLS has a single-dimensional structure and is negatively correlated with anxiety and depression. What are the substantial implications for assessing the quality of life among elderly in Peru

2How does this study providing a reliable measurement tool for life satisfaction among elderly in Peru impact future research and policy-making?

3Can the findings of this study be generalized to other Latin American countries? Why or why not?

I hope the author can use these questions to further refine the paper and improve the quality of the research. Thank you very much!

Author Response

  1. Con el aumento de la población de personas mayores en América Latina, ¿por qué se ha vuelto crucial centrarse en la satisfacción con la vida de las personas mayores?

Agradecemos la observación realizada. Se incluye información que respalda el aumento de la población de adultos mayores y la importancia de la satisfacción con la vida durante el envejecimiento. Se agregó lo siguiente: “Estos factores presentan desafíos para los servicios sociales y de salud en la búsqueda de una vida satisfactoria [9]. Por lo tanto, mejorar la salud y la satisfacción con la vida de los adultos mayores se vuelve crucial, ya que estar contento con la vida actual se considera un indicador de envejecimiento saludable [10]. Adicionalmente, la satisfacción con la vida, al ser una evaluación subjetiva, es susceptible a cambios contextuales y se ve influenciada por las percepciones e interpretaciones de los adultos mayores, relacionadas con su salud y las condiciones sociales y económicas de su entorno [11]. Al respecto, Diener [12] enfatiza la importancia de la evaluación cognitiva global que los adultos mayores hacen de sus propias vidas. Diversos estudios indican una relación positiva entre la satisfacción con la vida y la salud subjetiva, así como su impacto en la autoestima, la motivación, la salud general y mejores estrategias de afrontamiento [13; 14]. En este contexto, la satisfacción con la vida cobra relevancia porque ayuda a desmitificar los estigmas sobre los adultos mayores.

  1. Al estudiar la satisfacción con la vida de los adultos mayores en Perú, ¿qué factores sociales y culturales específicos deben considerarse?

Agradecido por la observación realizada. Se ha incluido información sobre los factores sociales y culturales a considerar para la satisfacción con la vida en los adultos mayores. Se agregó lo siguiente:: El Perú también refleja esta tendencia demográfica; según el Instituto Nacional de Estadística e Informática [INEI] [5], las personas mayores de 60 años representan el 13,6% de la población del país, y se estima que para el 2070, la proporción de adultos mayores alcanzará el 30,1%. En cuanto a los factores sociales, el 33,8% de los adultos mayores que trabajan están afiliados a un sistema de pensiones, y el 26,8% de los hogares peruanos tienen al menos un adulto mayor como jefe de hogar [5]. Por lo tanto, la calidad de vida de los adultos mayores requiere de la continuidad del empleo hasta edades muy avanzadas, y la calidad de las redes de apoyo también es relevante ya que puede ser un factor clave en la satisfacción con la vida. En salud, a pesar de que el 92,1% de las personas mayores de 60 años cuentan con algún tipo de seguro de salud, existe una tendencia decreciente en la búsqueda de atención médica debido a la alta demanda y a los ineficientes procesos de atención [6]. Estas problemáticas han llevado a los adultos mayores a experimentar depresión, ansiedad, soledad y sentimientos de abandono, afectando significativamente su calidad de vida [7]. Entre los factores culturales, los adultos mayores valoran la salud funcional a través de la independencia y la capacidad de realizar actividades cotidianas; también aprecian la participación activa en actividades religiosas, de ocio y de integración social [8].           

  1. What is the current research status regarding elderly quality of life and satisfaction in Latin America, particularly in Peru?

The information regarding the need to assess life satisfaction in Latin America was improved. The following was added: In Latin America, the SWLS has been validated in older adults in countries such as Mexico [27], Peru [28], Chile, and Ecuador [29], focusing on exploratory, confirmatory, and invariance methods. Given that the SWLS is widely used, new validations con-sidering cultural variation are necessary, as differences and similarities in life satisfac-tion judgments emerge over time due to sociocultural influences [30].

  1. What innovations does Exploratory Graph Analysis (EGA) offer compared to traditional factor analysis? How does it enhance the understanding of the structure of the SWLS?

The comparison between EGA and traditional factor analysis was specified more precisely. The following was added: An innovative way to address the relationships between items is through network analysis [33; 34]. One of the challenges is the ability to visualize relationships in a dia-gram composed of nodes (items) and edges (partial correlations); additionally, the thickness of the edges allows for the examination of the strength of the relationships [35]. Exploratory Graph Analysis (EGA) is combined with a set of weighted networks [36], enabling the examination of network loadings in the node diagram, structural consistency, and facet detection algorithms [34]. In this way, EGA allows for the im-mediate interpretation of elements belonging to each factor through the network graph using colors, and influential relationships between items and dimensions can be ob-served without the need to make decisions about the type of rotation to use for the factor structure [37]. Therefore, EGA is a useful tool for exploring the factor structure and item interactions of the SWLS.

  1. Why choose EGA over traditional factor analysis to analyze the dimensional structure of SWLS?

The comparison between EGA and traditional factor analysis was specified more precisely. The following was added:In the last decade, network models have emerged as an alternative for exploring data structures; these models also complement existing latent variable techniques such as multidimensional scaling and exploratory factor analysis [31]. In contrast to latent variable models derived from Classical Test Theory (CTT), the determination of the internal structure of latent factors can lead to a lack of consensus in the definition and interpretation of the obtained factors [32].

  1. What challenges did the article encounter in applying the EGA method to analyze SWLS?

The challenges presented by the EGA method for analyzing the SWLS were specified more precisely. The following was included: One of the challenges is the ability to visualize relationships in a diagram composed of nodes (items) and edges (partial correlations); additionally, the thickness of the edges allows for the examination of the strength of the relationships [35]. Exploratory Graph Analysis (EGA) is combined with a set of weighted networks [36], enabling the exam-ination of network loadings in the node diagram, structural consistency, and facet de-tection algorithms [34]. In this way, EGA allows for the immediate interpretation of elements belonging to each factor through the network graph using colors, and influ-ential relationships between items and dimensions can be observed without the need to make decisions about the type of rotation to use for the factor structure [37]. Therefore, EGA is a useful tool for exploring the factor structure and item interactions of the SWLS.

  1. The study recruited 407 elderly aged 60 to 95 using non-probability convenience sampling. What are the advantages and disadvantages of this method?

The advantages and disadvantages of the non-probabilistic sampling method were specified more precisely. The following was included: Among the advantages of this type of sampling are the adaptability of the design when the sample is specific and difficult to reach, the estimations can be sufficiently accurate if applied correctly, and it is useful for obtaining preliminary data. However, the dis-advantages of non-probabilistic sampling, particularly concerning representativeness and generalizability, must be noted [41].

  1. How was the accuracy and reliability of data ensured when collecting data using SWLS, PHQ-9, and GAD-7 scales?

To ensure the accuracy and reliability of the information, it was verified that the measures had psychometric evidence, based on the studies by Cjuno et al. [43], Franco-Jimenez and Nuñez-Magallanes [44], and Caycho-Rodriguez et al. [28]. For this study, the internal consistency reliability of the three instruments was also calculated, with the PHQ-9 scale obtaining a coefficient of 0.89, the GAD-7 a coefficient of 0.92, and the SWLS a reliability coefficient of 0.92.

  1. What was the process of data collection like in Peru for such a survey study? Were there difficulties in obtaining data?

The data collection process for this study followed these steps: first, four Comprehensive Senior Centers (CIAM) in the districts of Lima were contacted. Second, the coordinators of these centers were informed about the study, and potential dates for administering the instruments were scheduled. Third, the researchers attended the CIAM during the seniors' activity sessions and preliminarily assessed their availability to participate in the study. Fourth, informed consent was provided, outlining the study’s objectives, confidentiality, anonymity, and voluntary participation.

Entre los retos encontrados, la administración de los instrumentos se realizó de manera individual. En el caso de los adultos mayores sin educación formal, se les leyeron en voz alta los enunciados y las opciones de respuesta.

Reviewer 2 Report

Comments and Suggestions for Authors

Dear Editors and Authors, thank you for the opportunity to review this study that explored the new psychometric evidence of the life satisfaction scale in older adults in Peru. The manuscript could be further improved in the clarity and readability. Some comments for the authors to consider. Hopefully, it can be helpful for further improvement.

Major comments:

Introduction: The rationale for conducting this study should be emphasized further.

Methods: The methodology section might be confusing; I defer to other reviewers for more detailed feedback.

Results: The results could also be further polished to improve the readability.

Other comments:

Line 16: What is SWLS? Please write out the full term.

Line 94: Table 1: Could I ask any reasons for why no education and primary education groups are obviously under-represented?  

Lines 111-124: Could I ask how these PHQ-9 and GAD-7 were linked to SWLS?

Lines 175-176: Table 2: It might be better to add captions for SWLS items 1-5 means for at the bottom of this table. Please also add explanations for the response options 1-5 means at the bottom of this table.

Lines 177-183: I might not quite understand the interpretation of the results. I would leave to other reviewers for their insights.

Line 194: Figure 1 Please add caption to explain the figure.

Line 213: Figure 2 Please add caption to explain the figure. 

Comments on the Quality of English Language

The quality of English could be further improved in both clarity and readability.  

Author Response

  1. Introduction: The rationale for conducting this study should be emphasized further.

Thank you very much for the feedback; the following information was included: The SWLS has been used in more than 4,000 studies to assess an individual's overall evaluation of their own life. This instrument is highly relevant and applicable, as the information provided through Exploratory Graph Analysis (EGA) can have significant implications for clinical practice with older adults. Understanding the dimensionality of the SWLS ensures that the scores derived from a single measure are valid and useful for assessing life satisfaction. Therefore, the SWLS can be a valuable tool for profes-sionals in psychology, psychiatry, and geriatrics, enabling specific screening in mental health settings.

  1. Methods: The methodology section might be confusing; I defer to other reviewers for more detailed feedback.

Without modifications, as no specific observations from the reviewers were found.

  1. Results: The results could also be further polished to improve the readability.

The presentation of the results was revised for better readability. A more comprehensive version of the notes was included at the bottom of Table 1. Additionally, the title and numbering of the figures were placed above the images.

Other comments:

Line 16: What is SWLS? Please write out the full term.

It was confirmed that the full name of the scale is mentioned at the beginning, stating: 'The Satisfaction with Life Scale (SWLS) was developed by Diener et al. [25]'. After this, the abbreviation SWLS was used consistently throughout the manuscript.

Line 94: Table 1: Could I ask any reasons for why no education and primary education groups are obviously under-represented?

For a better representation of the subgroups in the educational level, it was reconsidered to categorize them as follows: No education, basic education, and higher education. In this process, older adults who were previously categorized under initial education and secondary education were grouped together under 'basic education.' Details of the revised table are provided below:

Table 1. Characteristics of Older Adults (n = 407)

Variable

Category

Frequency

%

Age (M ± SD)

69.5 ± 6.7

Sex

Male

160

39.3

Female

247

60.7

Living Arrangement

Lives alone

35

8.6

Lives with spouse

106

26

Lives with children

120

29.5

Lives with spouse and children

112

27.5

Marital Status

Single

40

9.8

Married

227

55.8

Cohabiting

26

6.4

Divorced/Separated

35

8.6

Widowed

79

19.4

Educational Level

No education

8

2

With basic education

181

44.4

Higher education

122

30

Employment

Yes

103

25.3

No

304

74.7

Physical Activity

Never

16

3.9

Rarely

78

19.2

Sometimes

177

43.5

Frequently

108

26.5

Very frequently

28

6.9

Lines 111-124: Could I ask how these PHQ-9 and GAD-7 were linked to SWLS?

The PHQ-9 and GAD-7 measures were used to establish evidence based on relationships with other variables. To clarify the instruments section more effectively, the following was added: 'Measures used for evidence based on relationships with other variables.

Lines 175-176: Table 2: It might be better to add captions for SWLS items 1-5 means for at the bottom of this table. Please also add explanations for the response options 1-5 means at the bottom of this table.

At the end of Table 2, a note was added explaining the meanings of the response options for items 1 to 5; the following was included: '1 = Strongly disagree; 2 = Disagree; 3 = Neither agree nor disagree; 4 = Agree; 5 = Strongly agree.

Lines 177-183: I might not quite understand the interpretation of the results. I would leave to other reviewers for their insights.

I understand the complexity of the analysis and offer an interpretative overview of the results that may help clarify their understanding: 'Figure 1 shows the network structure of the SWLS items using EGA, where a single dimension is evident. Network loadings (nodes) were examined, revealing high values (>0.35) for most items, except for item 1, which presented a moderate loading (>0.25). Regarding structural consistency, it was observed that in 100% of the cases, a single dimension was identified, with the items consistently grouped within one community'.

Line 194: Figure 1 Please add caption to explain the figure.

The title for Figure 1 was included at the top of the figure.

Line 213: Figure 2 Please add caption to explain the figure. 

The title for Figure 2 was included at the top of the figure.

Reviewer 3 Report

Comments and Suggestions for Authors

The current study presents a life satisfaction scale for older adults, adapted and validated for Peruvian older adults. Older adults from care centers were recruited for the validation and the validity and reliability of the scale were analyzed using different statistical tests and models, proving the instrument's robustness and value for further research studies

 The manuscript could benefit from some improvements:

1. Provide data regarding older population and life expectancy from 2024 and not from 2022. Distinguish between life expectancy and health life expectancy – it is relevant for the current study, since participants were recruited from different care centers.

2. Use the definition of life expectancy agreed by the WHO, UN or other international forums. The same regarding healthy life expectancy

3. It is not clear how life expectancy and healthy aging are related concepts and influence one another. Authors claim that life expectancy is a “component of healthy aging”. Elaborate on this – it is not intuitive how life expectancy is a component of healthy aging. Also, how life expectancy is linked to good health and social engagements. Healthy life expectancy is a different concept and  even for this concept it is not clear how it will be a component of healthy ageing (a concept which discusses different other aspects of the older people’s life)

4. What is the relationship between life expectancy and subjective well-being? Both terms are used in the text, but the relationship between them is not explained

5. Some factors are indicated as influencing life satisfaction. There are numerous others. Why are only those factors mentioned in the text? Which are the criteria for selecting exactly those factors?

6. There is a lot of relevant literature on life satisfaction and older adults:  concept, operationalization, and measurements, factors (antecedents and consequences). This literature is not presented in the paper and the literature review part of the current manuscript needs to be seriously revised.

Comments on the Quality of English Language

Author Response

1. Provide data regarding older population and life expectancy from 2024 and not from 2022. Distinguish between life expectancy and health life expectancy – it is relevant for the current study, since participants were recruited from different care centers.

The text was adjusted according to the suggested revisions: The increase in life expectancy among older adults (OAs) has driven the devel-opment of new social, economic, and health policies aimed at improving quality of life [1] and has also led to an increase in scientific research [2]. While demographic projec-tions show a rise in the average life expectancy of older individuals, healthy life ex-pectancy refers to the number of years of good health that an individual can expect to live at a given age [3].

2. Use the definition of life expectancy agreed by the WHO, UN or other international forums. The same regarding healthy life expectancy

The definition of healthy life expectancy was adjusted as follows: 'While demographic projections show an increase in the average life expectancy of older adults, healthy life expectancy refers to the number of years of good health an individual can expect to live at a given age [3] '.

3. It is not clear how life expectancy and healthy aging are related concepts and influence one another. Authors claim that life expectancy is a “component of healthy aging”. Elaborate on this – it is not intuitive how life expectancy is a component of healthy aging. Also, how life expectancy is linked to good health and social engagements. Healthy life expectancy is a different concept and  even for this concept it is not clear how it will be a component of healthy ageing (a concept which discusses different other aspects of the older people’s life)

Thank you for the observation. For this section, an explanation of the relationship between life satisfaction and aging was considered, and the section was improved to enhance the clarity of the text: “Therefore, improving the health and life satisfaction of older adults becomes crucial, as being content with one’s current life is considered an indicator of healthy aging [10]. Additionally, life satisfaction, being a subjective assessment, is susceptible to contextual changes and is influenced by the perceptions and interpretations of older adults, relating to their health and the social and economic conditions of their environment [11]. In this regard, Diener [12] emphasizes the importance of the global cognitive evaluation that older adults make of their own lives”.

4. What is the relationship between life expectancy and subjective well-being? Both terms are used in the text, but the relationship between them is not explained

Thank you for the feedback. The entire text was revised to now explain the relationship between life satisfaction and healthy aging.

5. Some factors are indicated as influencing life satisfaction. There are numerous others. Why are only those factors mentioned in the text? Which are the criteria for selecting exactly those factors?

Grateful for the observation made. Information about the social and cultural factors to consider for life satisfaction in older adults was included. The following was added: “Peru also reflects this demographic trend; according to the National Institute of Statistics and Informatics [INEI] [5], people over 60 years old make up 13.6% of the country's population, and it is estimated that by 2070, the proportion of older adults will reach 30.1%. Regarding social factors, 33.8% of working older adults are affiliated with a pension system, and 26.8% of Peruvian households have at least one older adult as the head of the household [5]. Therefore, the quality of life for older adults requires continued employment until very advanced ages, and the quality of support networks is also relevant as it can be a key factor in life satisfaction. In health, despite 92.1% of people over 60 years old having some form of health insurance, there is a decreasing trend in seeking medical care due to high demand and inefficient care processes [6]. These issues have led older adults to experience depression, anxiety, loneliness, and feelings of abandonment, significantly affecting their quality of life [7]. Among cultural factors, older adults value functional health through independence and the ability to perform daily activities; they also appreciate active participation in religious, leisure, and social integration activities [8].

Round 2

Reviewer 2 Report

Comments and Suggestions for Authors

Dear Editors and Authors, thank you for the opportunity to review this study that explored the new psychometric evidence of the life satisfaction scale in older adults in Peru. Minor comments for the authors to consider. 

Line 305: Is "M" referring to the mean or median?

Table 1: Please add annotations for "M" and "SD," as you have done in Table 2.

Line 380: ‘read aloud to them’ for …??

Table 2: Please revise the SWLS items to use more readable language, or alternatively, provide annotations explaining SWLS-1 through SWLS-5.

Comments on the Quality of English Language

Further improvements can be made in the academic writing and precision. 

Author Response

Response to the reviewer

1. Line 305: Is "M" referring to the mean or median? Table 1: Please add annotations for "M" and "SD," as you have done in Table 2.

Thank you very much for the observation. A note has been added to Table 1 indicating that 'M = Mean' and 'SD = Standard Deviation'.

2. Line 380: ‘read aloud to them’ for …??

The wording of the process was revised to improve the coherence and clarity of the data collection. Therefore, the following was included: Fifth, once the older adults agreed to participate, they completed the questionnaires individually. For those who were illiterate, the statements and response options were read aloud to ensure accurate completion of the instruments, thereby avoiding biases related to reading and comprehension difficulties.

3. Table 2: Please revise the SWLS items to use more readable language, or alternatively, provide annotations explaining SWLS-1 through SWLS-5.

We appreciate you highlighting this observation. Consequently, a note was added at the bottom of the table with the description of the SWLS item: SWLS – 1 = In most respects, my life is as I want it to be; SWLS – 2 = The conditions of my life are good; SWLS – 3 = I am satisfied with my life; SWLS – 4 = So far I have gotten the important things I want in life; SWLS – 5 = If I could live my life over, I would repeat it just the same way it has been.

4. Further improvements can be made in the academic writing and precision.

Thank you very much for the observation. We have carefully reviewed the text to ensure it meets the fundamental criteria for writing and grammar. We will remain attentive to any further inaccuracies that may arise in our manuscript.

Reviewer 3 Report

Comments and Suggestions for Authors

The manuscript has improved  compare to the last version and it has the merits to be accepted  by the journal.

Author Response

We sincerely appreciate your time and academic expertise, which have been crucial in improving our manuscript. Please find the final version attached.

Regards,

Julio Dominguez-Vergara
